# Quantitative examination of the anatomy of the juvenile sugar maple xylem

**Tenaya Driller[1], James A. Robinson[1], Mike Clearwater[2], Daniel J. Holland[1], Abby van den Berg[3], Matthew Watson[1] \***

1 Biomolecular Interaction Centre & Department of Chemical and Process Engineering, University of Canterbury, Christchurch, New Zealand, 2 School of Science, University of Waikato, Hamilton, New Zealand, 3 Proctor Maple Research Center, University of Vermont, Underhill, Vermont, United States of America

\* matthew.watson@canterbury.ac.nz

**Data Availability Statement:** The data underlying the results presented in this study are available from Open Science Framework. We have uploaded

## Abstract

New methodologies have enabled viable sap yields from juvenile sugar maple trees. To further improve yields, a better understanding of sap exudation is required. To achieve this, the anatomy of the xylem must first be fully characterised. We examine juvenile maple saplings using light optical microscopy (LOM) and scanning electron microscopy (SEM), looking at sections cut along differing orientations as well as macerations. From this we measure various cell parameters. We find diameter and length of vessel elements to be 28 ± 8 μm and 200 ± 50 μm, for fibre cells 8 ± 3 μm and 400 ± 100 μm, and for ray parenchyma cells 8 ± 2 μm and 50 ± 20 μm. We also examine pitting present on different cell types. On vessel elements we observe elliptical bordered pits connecting to other vessel elements (with major axis of 2.1 ± 0.7 μm and minor 1.3 ± 0.3 μm) and pits connecting to ray parenchyma (with major axis of 4 ± 2 μm and minor 2.0 ± 0.7 μm). We observe two distinct pit sizes on fibres with circular pits 0.7 ± 0.2 μm in diameter and ellipsoidal pits 1.6 ± 0.4 μm by 1.0 ± 0.3 μm. We do not observe distinct pitting patterns on different fibre types. The various cell and pit measurements obtained generally agree with the limited data available for mature trees, with the exception of vessel element and fibre length, both of which were significantly smaller than reported values.

## 1. Introduction

Sugar maple trees (*Acer saccharum* Marshall) have extraordinary ways in which they develop elevated stem pressures during freeze-thaw events. This leads to springtime sap exudation when xylem vessels are exposed to atmospheric pressures through wounding [1], with syrup producers often applying vacuum to further improve sap yields [2]. Although the exact mechanisms involved in this process have yet to be fully resolved, a maple tree's unique stem structure, specifically the xylem tissue, plays an instrumental role. The xylem provides structural support to the tree stem and is an important region of conduction and storage of water and non-structural carbohydrates within the tree. In sugar maple trees, the secondary xylem makes up the majority of the stem [3], and behaviour occurring within this xylem tissue during

our data to the open science framework at the following doi:10.17605/OSF.IO/QKBNM

**Funding:** This research was funded via a seed grants from the Biomolecular Interaction Centre and a seed grant from the University of Canterbury Faculty of Engineering, both awarded to MW. Funding was also provided from the New Zealand Ministry of Business Innovation and Employment's (MBIE's) Smart Ideas Research Project under contract UOCX2110 awarded to MW. The funders had no role in study design, data collection and analysis, decision to publish, or preparation of the manuscript

**Competing interests:** The authors have declared that no competing interests exist.

freezing is believed to result in their ability to develop elevated stem pressures while in a leaf-less state [1].

Information on the elements within the xylem is necessary in order to fully understand the function of each cell type and role they might play in sap exudation. Knowledge of the size, distribution and cell wall structure of each xylem element is needed to accurately understand the tree stem's hydraulic conductivity, patterns of sap flow and the other important factors involved in maple sap exudation such as capillary and osmotic forces. The composition of cell walls, and pitting along each, provides valuable information on whether sucrose molecules are prevented from traveling from vessel elements to the fibres, as hypothesized in the osmotic theory of maple sap exudation [1]. Furthermore, an understanding of the pitting along xylem walls is useful for identification purposes in wood anatomical studies, understanding the functionality of the xylem elements in terms of water conduction, and providing means to differentiate between fibre types.

There have been a handful of attempts to model the mechanisms of sap exudation [4–6]. In order to develop accurate models for prediction of sap exudation and overall sap yields there is a need for detailed information on xylem structure. Currently there are only a few works providing different anatomical data for mature maple trees [7–15].

Traditional maple syrup production extracts sap from mature maple trees via tapping. However, recent developments in sap extraction methodologies has allowed for (in theory) commercially viable maple syrup yields from small diameter maple saplings (5–10 cm) [16, 17]. This offers up potential new regions for syrup production, as small diameter saplings mean less extreme freeze thaw cycles may be required to induce sap flow, opening up production to warmer climates [18]. However, there is currently a lack of detailed anatomical information on xylem elements within juvenile sugar maple. For this reason, there is significant motivation to examine saplings.

In this work we utilise light optical microscopy (LOM) and scanning electron microscopy (SEM) to study the structure of the juvenile sugar maple stem, with a focus on the aspects of the microstructure that will provide insight into the mechanisms involved in maple sap exudation. The use of micrographs of transverse, tangential, radial and macerated samples to measure cell properties is well established [19, 20]. SEM and LOM are used as they are accessible, and extremely versatile microscopes that offer a simple and inexpensive way to acquire basic structural information within maple trees. We aim to supplement the small amount of information available on the anatomy of sugar maple trees and provide direct measurement of factors necessary to characterise the hydraulic connections between various cellular structures. Where possible we endeavour to provide comparison to existing data on mature trees, though such data are limited.

## 2. Methods

Plant material for experiments using LOM and SEM was taken from branch and stem sections of sugar maple saplings. Five saplings, 1–4 years old, were sourced from local nurseries and kept at the University of Canterbury for a two year period. During this time saplings were fertilized, well-watered and kept outside in 20 litre pots. Branch and/or stem sections 3–15 mm in diameter were taken from each sapling for analysis across the two years. An additional 6 saplings, 1–2 years old, were purchased from nurseries. LOM/SEM images were also taken from samples of these saplings. These images were only used in analysis of vessel diameter.

Samples were prepared for imaging on standard LOM and SEM equipment. LOM samples were either embedded in paraffin wax, or preserved in ethanol, before being sectioned using a microtome (a Leica RM2165 for embedded samples and a Rechert Om E for non-embedded

samples) to produce samples 30–40 μm thick (for non-embedded samples) or 12 μm thick (for embedded samples). LOM samples were either stained with Safranin or Safranin and Fast green, following Donald Alexander Johansen [21] or with Safranin and Astra Blue, following Vazquez-Cooz and Meyer [22]. Staining with Safranin and Astra Blue was only used for identifying different fibre types (as outlined in section 3.A.IV) and unless otherwise stated all LOM images show samples stained with Safranin. Images were analysed identically regardless of the staining used. SEM samples were cut to create a flat surface, dehydrated in alcohol, and sputter coated with either gold or platinum (using a Quorum EMS 150T ES). Section views were produced, as shown in **Fig 1**. In addition to sections, macerations were also prepared according to Franklin [23], and imaged using LOM and SEM (see **Fig 2**).

Images were analysed using Fiji [24]. Fibre and vessel diameters were measured on >40 images, and cell diameters on >20. Other observations were based on 2–10 images. The cell diameters given in Section 3.A were evaluated from sectioned samples, using a manually chosen threshold for each set of images to segment cells, and particle analysis tools used to identify and measure shape descriptors for each cell. The cell diameters given are the equivalent circle diameter and for all measurements one standard deviation is reported as a measure of the variability. Lengths of vessel elements and fibres were measured using line tools on macerations, as shown in **Fig 2**. Ray parenchyma (and individual ray parenchyma cell) dimensions were measured similarly on sectioned samples.

Cell pitting was examined using different sections and macerated samples, with the size of pits manually measured using the 'ellipse' tool in ImageJ. We report either the dimensions of the ellipse, where pits were elliptical, or the equivalent area circle diameter, where they were circular.

## 3. Results and discussion

### 3.A. Cell examination

**3.A.I. Ray parenchyma.** In our images, ray parenchyma are easily identified by the fact they are arranged perpendicular to other xylem elements, and because of the presence of starch within the ray cells (see **Fig 3(B) and 3(C)**).

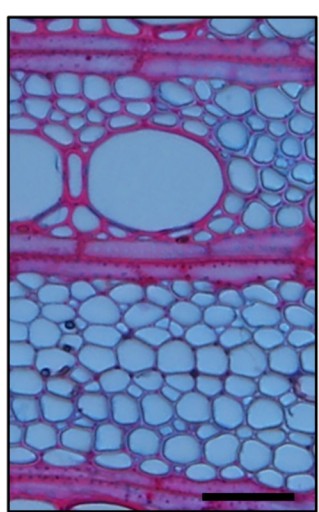 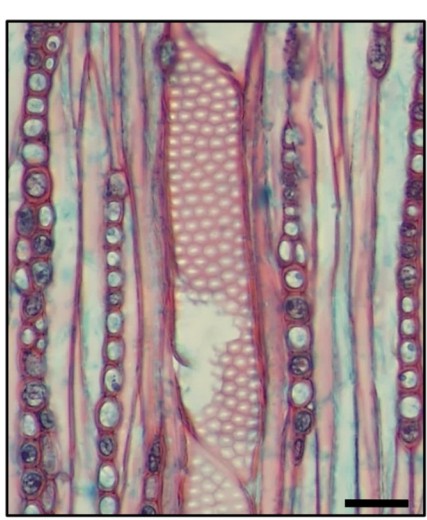 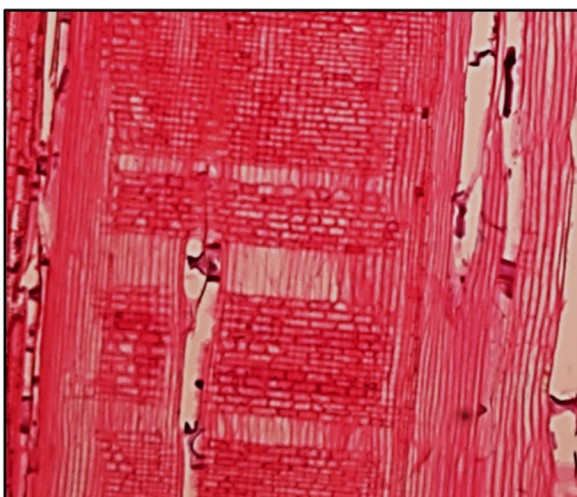

TRANSVERSE   TANGENTIAL   RADIAL

**Fig 1. LOM images showing maple xylem from different perspectives.** Scalebar is 25 μm for left and middle image, and 50 μm for right image.

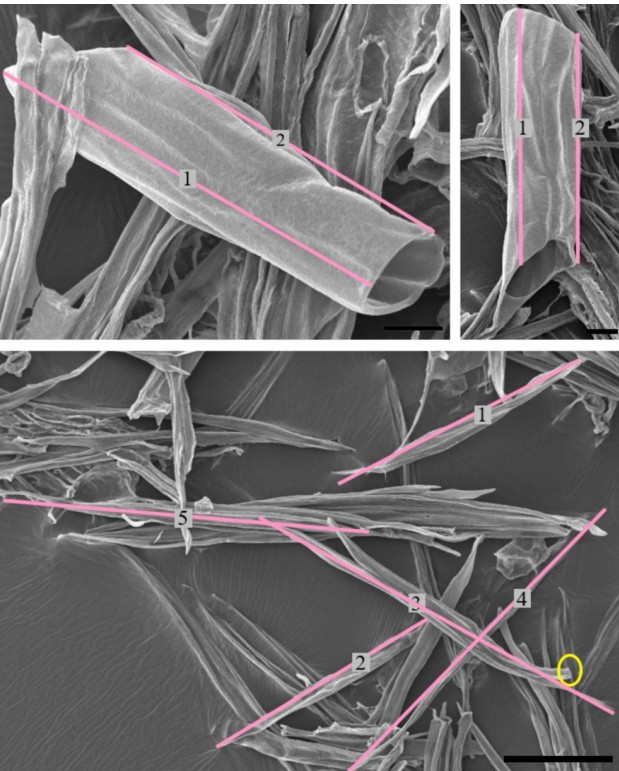

**Fig 2. Macerated xylem cells observed using SEM.** Lines used to measure the cell lengths are shown in pink. The yellow circle indicated a fibre which was broken and not measured. The scale bar in the top images is 25 μm while the bottom is 100 μm.

Fig 4 shows the distribution of the number of cells in each of the rays measured. Ray parenchyma cells were found to have an average diameter of 8 ± 2 μm, and length of 50 ± 20 μm. As Fig 4A shows, the majority of rays observed were uniseriate, though instances of ray parenchyma up to five cells wide were observed, which agrees with previous work [7]. Fig 4B shows that most of the rays were about ten cells long though ray height ranged from 2–59 cells, with the highest frequency range of the histogram being 6–10 cells, and the average height being 200 ± 100 μm. A prior study reported multiseriate rays in mature trees reaching heights of 800 μm and uniseriate rays less than 200 μm [8]. The width and height of rays has been shown to fluctuate largely across the same species and is highly dependent on location within a tree [25].

The average radial length of a ray was 1.7 ± 0.6 mm, this was in branch or stem sections that were typically just over 3 mm in diameter, meaning the rays reached the entire distance from cambium to the pith. We expect this to be a result of the saplings' young age. As the trees age and the branch or stem sections grow radially, it is likely individual rays would no longer cover the entire distance from cambium to the pith. We are unaware of any existing information on the radial length of rays in mature sugar maple trees to compare.

The distance between rays remained roughly the same throughout the xylem region. An average distance of 70 ± 30 μm lay between rays, occupied by other cell types.

**3.A.II Axial parenchyma.**   Axial parenchyma were identified in transverse sections by their highly lignified cell wall and the presence of starch within the cells, both common features in the *Acer genus* [26]. They were also identified through their smaller size and elliptical shape compared with surrounding fibres. Pitting was also used to identify some axial parenchyma, which is discussed in section 3.B.I.

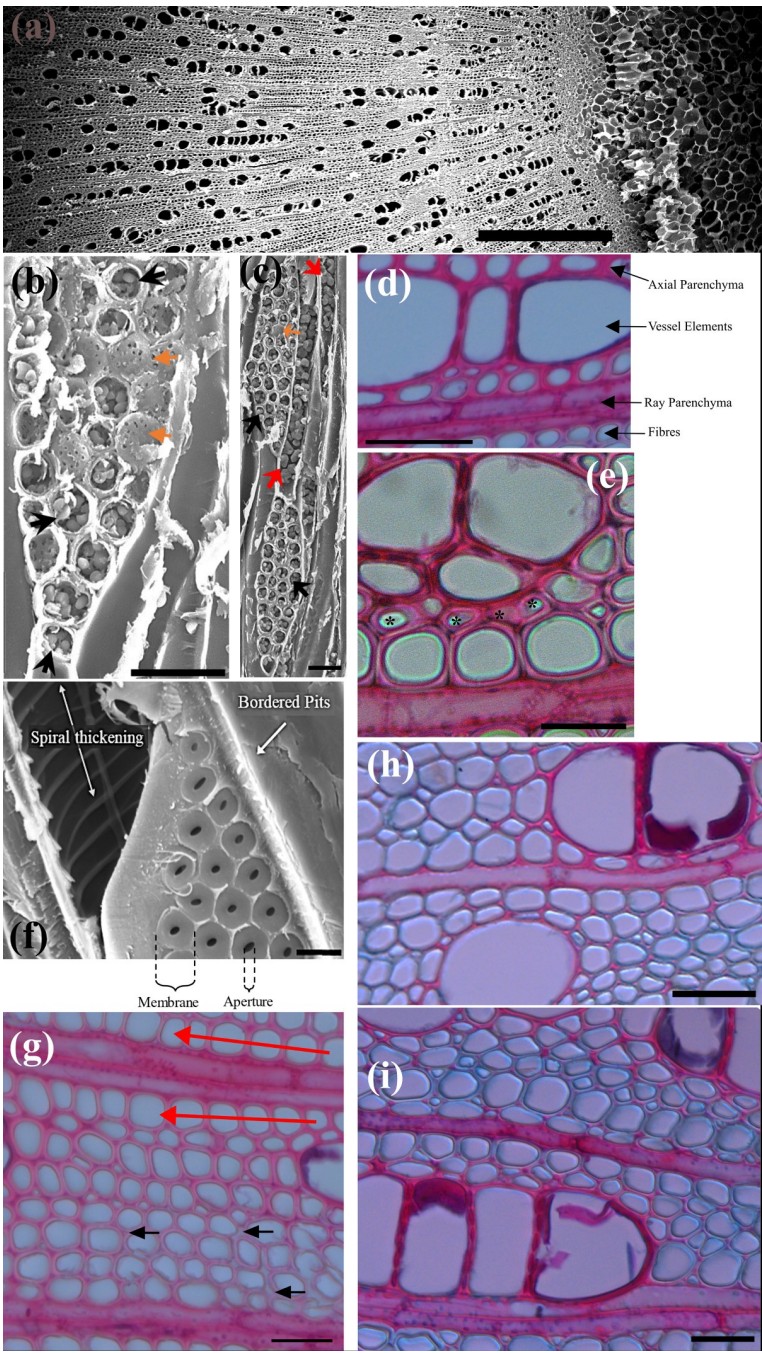

**Fig 3. LOM and SEM images of different xylem sections.** (a) zoomed out section of xylem (scale = 500 µm), (b-c) tangential xylem slice showing starch in ray parenchyma (black arrows) and axial parenchyma (red arrows). Ray Parenchyma pitting is shown (orange arrows). (d-e) transverse xylem sections showing cells believed to be axial parenchyma. (f) view of vessel element showing spiral thickening and pits (scale = 10 µm). (g) transverse xylem slice showing fibres. Black arrows point to intercellular spaces while red arrows point to radial files of larger fibres. (h-i) transverse xylem sections using safranin and astra blue [22]. Unless otherwise noted scale is 25 µm.

Cells which we believe to be axial parenchyma were observed in direct contact with vessels and/or ray parenchyma consistent with other studies on the *Acer genus* [11, 26]. We identified many instances of radial files of axial parenchyma between vessel elements and adjacent ray

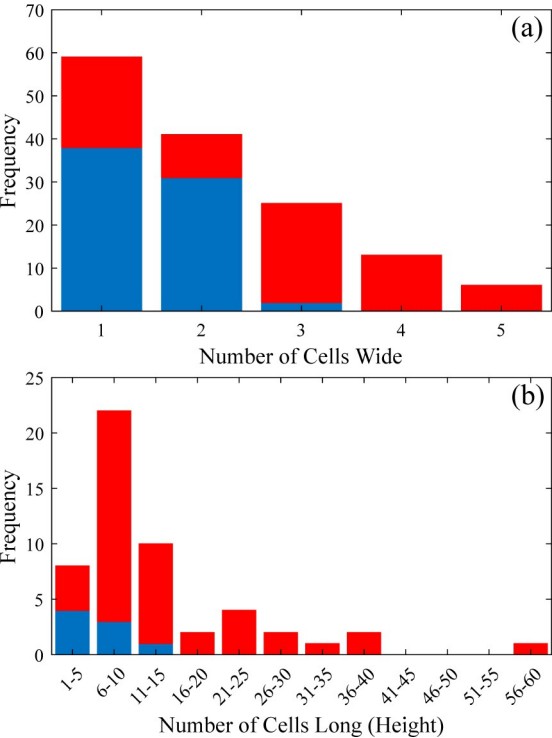

**Fig 4.** Histograms showing ray (a) width and (b) height in terms of no. of cells. For (a) 71 LOM measurements (in blue) and 73 SEM measurements (in red) were considered. For (b) it is based on analysis of 8 LOM and 44 SEM measurements.

parenchyma, as shown in **Fig 3D**, a feature not previously reported in sugar maple but which has been shown to occur in *Acer mono* [27].

In transverse sections the defining features of axial parenchyma were not always readily apparent (see **Fig 3E**), and fibres could easily be mistaken for axial parenchyma. The presence of starch is identifiable in SEM images (see **Fig 3C**), however in *Acer* there are also living fibres present that contain starch [28]. For these reasons we do not provide anatomical measurements for axial parenchyma. Further investigation would be required to enable reliable identification of axial parenchyma and living fibres.

**3.A.III. Vessels.** **Fig 2** shows examples of individual vessel elements imaged using macerated samples of xylem. They are easily identified by their large size and distinct tapered ends. The observed shape of vessel elements fluctuated considerably, relative to other vessel elements. We found short and wide vessel elements, and long, narrow vessel elements.

Most vessel elements were between 200 and 250 μm in length (**Fig 5**). However, lengths ranged from 91 to 320 μm with an average length of 200 ± 50 μm. Measurements from Panshin [8] reported an average length of 400 ± 90 μm for vessel elements in *Acer*, about twice our measured length measurements for sugar maple. The value from Panshin [8] was representative of the *Acer* genus as a whole. Comparing these values, our measured values could suggest that sugar maple trees have shorter vessel elements than the rest of the *Acer* genus, or the difference in length could be a result of measuring vessels from young saplings rather than mature trees.

**Figs 1** and **3A** show LOM images of a transverse xylem sections, showing the distribution of conduits, with vessels elements being identified by their large diameters. Most vessel elements were in direct contact with axial parenchyma, ray parenchyma and fibre tracheids with numerous pits connecting vessel elements to these other xylem elements. In transverse

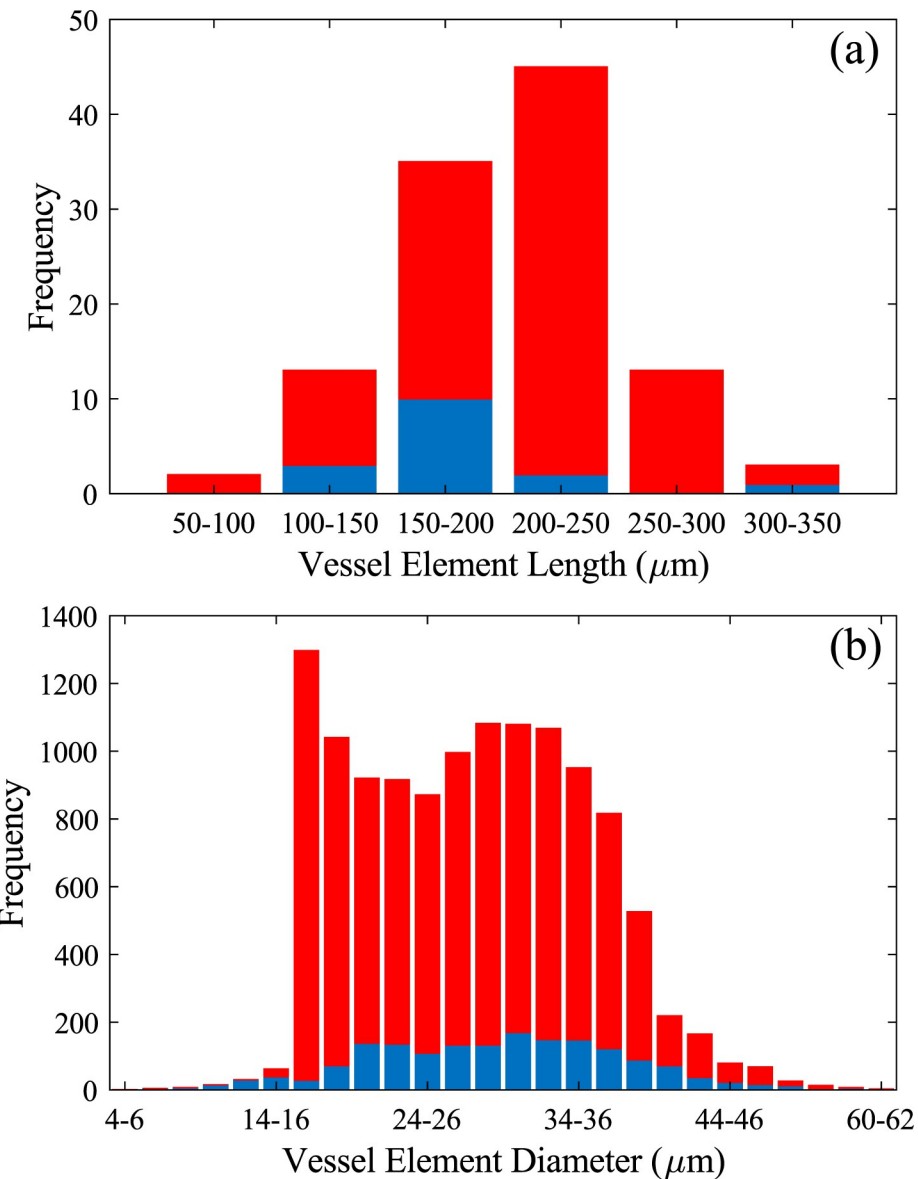

**Fig 5.** Histograms showing vessel element (a) length and (b) width. For (a) 16 LOM measurements (in blue) and 96 SEM measurements (in red) were considered. For (b) it is 1697 LOM and 10563 SEM measurements.

sections, vessel elements were found to exist alone, or in groups ranging from two to as many as eight. Vessel groups tended to extend along the direction of the ray parenchyma forming long radial files. From a transverse perspective, solitary vessels were elliptical in shape while those in groups were more often half-ellipses, with a shared straight wall between them.

Measurements of vessel diameter from transverse sections are summarised in **Fig 5B**. Vessel element diameters range from 6 µm to 55 µm, with an average diameter of 28 ± 8 µm. The highest frequency diameter range in the histogram was 16–18 µm. There is a considerable drop off in the number of vessels with diameters smaller than 16 µm, this is likely due to the limited ability of the image analysis technique to differentiate smaller vessels from fibres and axial parenchyma. Previous literature has reported sugar maple vessel element diameters

measuring 39 ± 7 µm [9] and conduit diameters can fluctuate greatly depending on their location within the tree, the surrounding climate and the individual tree itself [7, 29].

In *Acer*, the cell walls of vessel elements are highly lignified and have a prominent spiral thickening (see **Fig 3F**). We observed spiral thickening on dozens of macerated vessel elements. The spiral thickening was consistently spaced across the entire length of the vessels. We analysed the spiral thickening for 7 of the vessel elements observed and found the average thickness of the spiral thickening was 0.7 ± 0.2 µm with an average distance of 2.5 ± 0.6 µm between each spiral. This result agrees with prior work on sugar maple in which spiral thickenings were observed, spaced 3–4 µm apart [8, 10].

**3.A.IV. Fibres.**   Hardwood tree species typically have two different types of fibres: libriform fibres and fibre tracheids. Current fibre terminology considers both fibre types to be different sub-types of fibres in *Acer* [30]. The international association of wood anatomy defines libriform fibres as 'elongated, commonly thick-walled cells with simple pits', and fibre tracheids as 'commonly thick-walled cells with a small lumen, pointed ends and bordered pit pairs' [31]. Although these distinctions hold for many angiosperm species, they are not as clearly defined in *Acer*. Many authors have reported difficulty in differentiating the fibre types based on pitting, stating that pits with distinct borders were not identified on any of the fibres observed [32, 33].

In *Acer*, including sugar maple, living libriform fibres are also present that, in addition to mechanical support, are thought to participate in secretion, transport, and storage [34]. However living fibres are almost identical to other fibres in size, shape, cell wall composition, and location within the xylem and so are difficult to differentiate by anything except based on the presence of starch [34, 35]. The existence of living fibres has been poorly documented in most species and the differences between living fibres and axial parenchyma are not well understood [34, 35].

**Fig 3H** shows a typical transverse image of fibres. There are differences in fibre size, and regions where fibres are closely packed, and regions where there are larger intercellular spaces. However, we cannot reliably distinguish different fibre types from such images, and as such measurement of fibre dimensions was done collectively.

Measurements are given in **Fig 6**. Fibres ranged from 213 to 1,001 µm in length with the majority being between 350 and 500 µm and the average length being 400 ± 100 µm. An average fibre length of 920 µm was reported by Panshin [8], about twice the length we recovered. This may suggest that maple saplings have shorter fibre lengths than mature trees, though further work would be required to confirm this.

Fibre diameters were measured on small regions of xylem. The results are presented in **Fig 6B**. Fibre dimensions followed a normal distribution with fibre diameters ranging from 2.53–22.28 µm and an average diameter of 8±3 µm. This agrees with the previously reported average fibre diameter of 7 µm in sugar maple stems [9]. In many cases, a row of larger-diameter fibres would run alongside sections of a ray. **Fig 3G** shows an example of this feature. This has not been previously noted in any literature on sugar maple saplings and was a common occurrence in our observations.

As noted, visually distinguishing between the two fibre types present in maple saplings was difficult. In our work, we were able partially separate the two separate fibre populations with the use of the differential staining technique described by [22]. Most cells types in the xylem, having cells walls with more lignin, will stain red, whilst libriform fibres, having less lignin, will stain blue.

Examples of the results of this staining are shown in **Fig 3H and 3I**. The results were mixed; some sections show a clear distinction between red and blue stained cells, while for others the difference between red and blue fibres is not clear. As can be seen in **Fig 3H and 3I**, vessel elements and most of the surrounding small cells appear a red colour, and the fibres further from

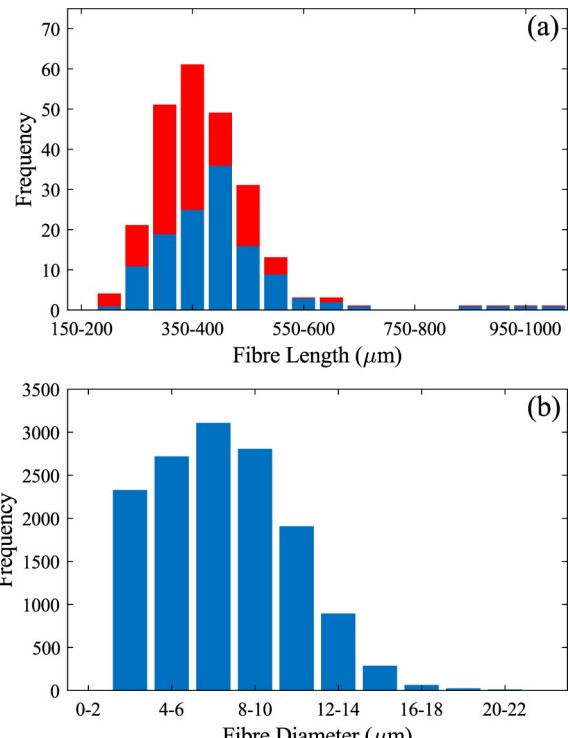

**Fig 6.** Histograms showing fibre (a) length and (b) width. For (a) 127 LOM measurements (in blue) and 114 SEM measurements (in red) were considered. For (b) it is 14105 LOM measurements.

vessels appear blue. Following Vazquez-Cooz and Meyer [22] these would be the fibre tracheids (red) and libriform fibres (blue) respectively. The distinction here is less apparent than in Vazquez-Cooz and Meyer [22], and fewer fibres are stained red. Staining is a complex process, factors such as the slice thickness, the specific batch of the stains, or the length of time the stain is left on can vastly affect the results observed. However, we believe this staining technique is promising and with refinement, can be potentially used to consistently distinguish different fibre types in sugar maple stems.

## 3.B. Cell pitting

**3.B.I. Ray parenchyma pitting.** Ray parenchyma cells showed similar pits to that observed for axial parenchyma cells and exhibited identical pitting patterns across all cell walls. We can observe tangential and transverse pitting in **Figs 3(B) and 3(C)** and **7A**, respectively. The average tangential pit diameter and transverse pit diameter were the same and measured to be 0.8 ± 0.2 μm. Due to the arrangement of ray parenchyma within sugar maple xylem, exposed transverse and tangential walls in which pits were measured would likely be bordering other parenchyma cells. Thus, one can assume all of the pits were parenchyma-parenchyma pitting. We were not able to obtain radial pitting measurements as when ray parenchyma were viewed from a radial perspective they were filled with contents, obstructing the view of the pits.

**3.B.II. Axial parenchyma pitting.** We found parenchyma cell walls contained numerous small, circular pits spread evenly across the transverse cell walls. These are visible in **Fig 7A** appearing as faint white dots along the pink cell wall of an axial parenchyma cell. All pits observed were of similar shape and size and had an average diameter of 0.8 ± 0.2 μm.

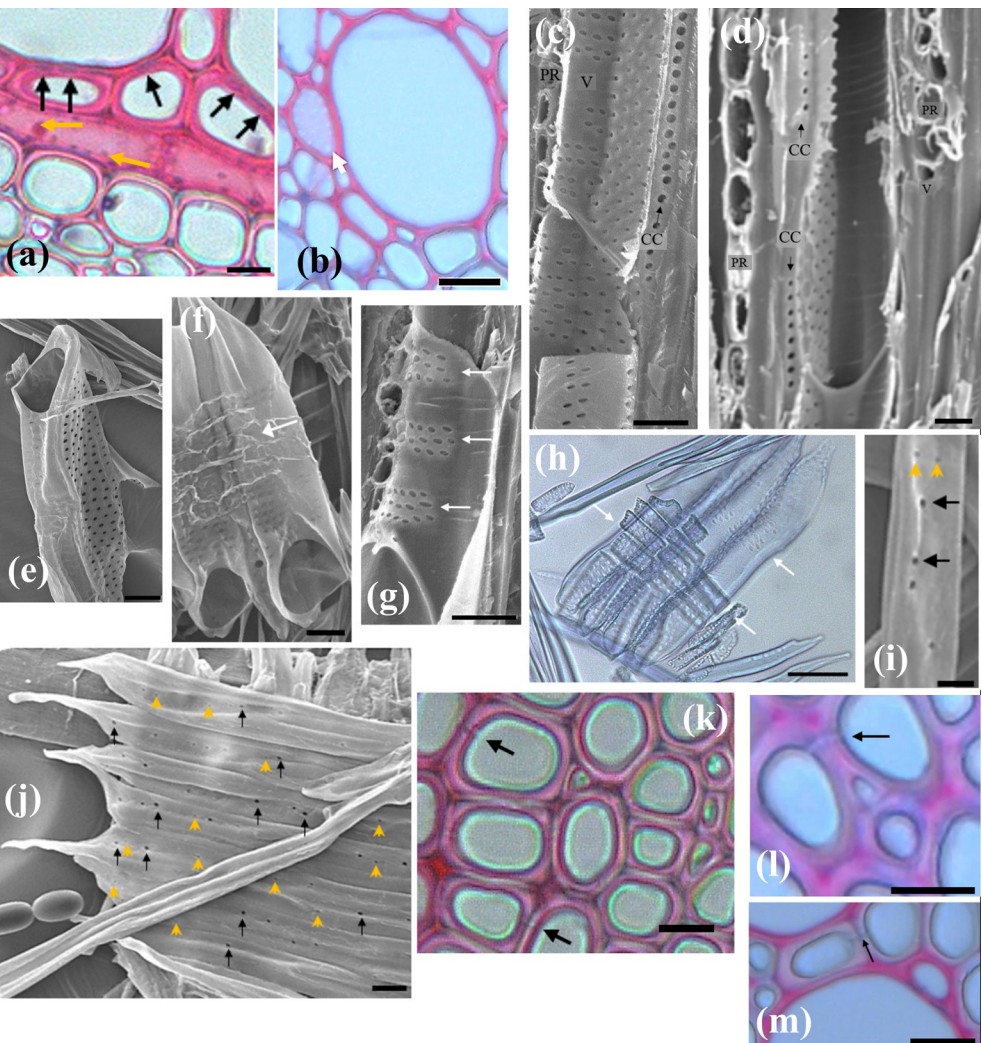

**Fig 7. SEM and LOM images of xylem sections and macerations.** (a-b) transverse xylem sections showing transverse axial parenchyma pitting (white arrow), transverse ray parenchyma pitting (orange arrow) and vessel-axial parenchyma bordered pit pairs (black arrows). (c-d) tangential xylem slices with ray parenchyma (PR) alongside vessels (V) and what we believe are contact cells (CC) identified by large circular pits (black arrows). (e) macerated vessel element (scale = 25 μm). (f-h) macerated xylem elements (scale = 20 μm) with rows of pits on vessel elements corresponding to connections to ray parenchyma (white arrows). (i-j) macerated fibre cells with small circular pits (yellow arrows) and larger elliptical pits (black arrows). (k-m) transverse xylem slices with black arrows pointing to examples of simple pits between fibres, some of which appear blind. Scalebar is 10 μm unless otherwise stated.

Connections between axial parenchyma and neighbouring cells were frequently observed. Vessel elements and axial parenchyma were connected through half-bordered pit pairs; a bordered pit on the vessel side, matched with a simple pit on the axial parenchyma cell wall. **Fig 7B** shows the vessel-axial parenchyma pitting connections from a transverse perspective. The pitting connections identified agree with earlier findings on *Acer mono* where half-bordered pit pairs were abundantly present between vessel elements and axial parenchyma [27]. We were unable to locate any pits between axial parenchyma and fibres. A prior study utilising SEM also noted that they did not find any pitting connections between fibres and adjacent axial parenchyma [11].

**3.B.III. Contact cells pitting.** There is little literature that exists on the size or shape of contact cells in *Acer*. Contact cells are supposedly most easily identified by their large elliptical pits (measuring 3–5 μm in diameter) on the walls adjacent to vessels [36].

In our experiments we identified structures with many large, circular pits. The pitting patterns observed were unlike any of the other pits on other cells and can be seen in **Fig 7(C) and 7(D)**. Pit apertures were large, with circular pits measuring 2.3 ± 0.7 μm diameter. Although the pits found were circular rather than elliptical, we believe these cells are contact cells due to the unique pitting pattern. Furthermore, we found very few instances of these cells occurring and always within close proximity to vessel elements, strengthening the case that they are contact cells.

**3.B.IV. Vessels pitting.** In our experiments, vessel pitting was most easily observed through macerated cells, as they provide an unobstructed view of each vessel element allowing us to measure the size and shape of the vessel pits. While this approach offers little information on the pitting connections, in the case of the vessel elements, imprints left behind from the cells that used to lie adjacent offer information on what used to exist there.

Vessel cell wall pitting seems to be extremely dependent on the type of cells lie adjacent. Some vessel walls were smooth and devoid of pits, while others exhibited quite pronounced bordered pits, tightly packed together in large groups. **Fig 7E** shows smooth vessel walls, likely where fibres ran adjacent, and heavily pitted walls, where another vessel element ran adjacent. The cell walls of vessel elements were also heavily pitted in instances in which ray parenchyma cells ran adjacent.

When vessel elements were found adjacent to each other, the walls between were almost entirely occupied by numerous bordered pits, packed tightly together, and spread across the entire vessel element length. A radial view showing vessel-vessel pitting can be seen in **Fig 3F** and a transverse perspective in **Fig 3(D) and 3(E)**. Vessel-vessel pits had elliptical apertures that measured 1.3 ± 0.3 μm and 2.1 ± 0.7 μm for the minor and major axes, respectively, and borders measuring 4.0 ± 0.6 μm in diameter. These borders were circular though appeared angular, likely through crowding, a feature described in earlier experiments on *Acer* by [8] who found pit borders in sugar maple to measure 10 μm in diameter.

Vessel-ray parenchyma pitting was commonly observed in macerated samples. These were identified by the remnants of ray parenchyma cells or small imprints of where ray parenchyma cells existed, running perpendicular to the vessel length. **Fig 7(F)–7(H)** shows examples of such occurrences, with pits occurring in horizontal bands across the vessel walls. Previous studies on sugar maple have reported profound pitting on vessel wall regions that lay adjacent to ray parenchyma cells through half-bordered pit pairs [26, 37, 38]. This connection is thought to facilitate the movement of starch and sugars between the parenchyma cells and the vessels.

The vessel-ray parenchyma pits identified along vessel cell walls were bordered with elliptical apertures measuring 4 ± 2 μm and 2.0 ± 0.7 μm for the major and minor diameters respectively. The vessel pits where ray parenchyma ran adjacent are similar in shape but slightly larger in diameter than the vessel-vessel pits. Vessel-ray parenchyma pitting measurements obtained agree with prior findings on sugar maple trees, in which the vessel cell wall pits are elliptical with the major and minor axes measuring 3.8 ± 0.6 and 2.5 ± 0.5 μm, respectively [11]. In previous studies, researchers have stated that half-bordered pit pairs were only observed between vessel elements and adjacent uniseriate rays, or uniseriate portions of multiseriate rays. Due to the nature of our experiments, we were unable to determine if this was the case for sugar maple.

**3.B.V. Fibre pitting.** The most common way to differentiate fibre types in hardwoods is through differences in their pitting, and the existence or extent of a pit border [39, 40]. However, in *Acer*, there has been controversy on how to distinguish fibre types based on pit size

and morphology [7]. Many authors do not recognize fibres with distinctly bordered pits in *Acer* [33, 41]. An early anatomical study on maple trees concluded that the fibres in maple were devoid of pits [26]. The observations of many subsequent researchers, however, have determined that this is not the case. Simple pits have been found scattered along libriform fibres, forming simple pit pairs between all libriform fibres [5, 11, 22, 40, 41]. The discrepancy between Wiegand [26] and subsequent studies likely has to do with the difficulty of properly observing pits along the cell walls, especially in the year 1906.

In sugar maple, fibre tracheids are almost always affiliated with vessels and connected through bordered pit pairs. Vessels, adjacent fibre tracheids, axial and ray parenchyma all appear well connected through bordered pits, while libriform fibres exhibit only blind pits, and most commonly no pits on walls adjacent to vessels [11]. There is general agreement that fibre tracheids, with their distinct pit borders, take part in water transport while libriform fibres serve a purely structural role as they lack pitting connections with all other xylem elements [10]. The tracheid-vessel pitting connection is well documented in other species [8, 27, 42], however was not reported in sugar maple trees until the work of Cirelli, Jagels [11]. Bordered pit pairs have been identified between fibre tracheids and vessel elements of maple trees [11].

The difference in shape and distribution of the pits along the fibre cell walls provide means to differentiate between the fibre types. Pit apertures of both types of fibres are elliptical, but the ends of the apertures of libriform fibre pits are apparently round and V-shaped toward the lumen [22]. In sugar maple, the bordered pits connecting fibre tracheids to vessels have distinct borders, measuring greater than 3 μm, and elliptical openings with an average diameter of 1.5 μm [22, 39]. Libriform fibre apertures were reported to be 0.78 ± 0.35 μm in diameter [11]. The pitting pattern in fibre tracheids is supposedly quite different to the other xylem cell types with longitudinally oriented groupings of bordered pits primarily on the walls adjacent to vessel elements [11]. Pits along fibre tracheid cell walls are supposedly spread evenly along the length of the cell, while libriform fibres pits are concentrated towards the centres [32].

In our experiments, we found fibre pitting was best observed with macerated cells. This perspective shows the distribution of pits along the length of the cell walls, but makes it difficult to determine adjacent elements. **Fig 7(I) and 7(J)** shows an example of macerated fibres with simple pits scattered along the length of the cell wall. We found two distinctly different pit types; small, circular pits and larger elliptical pits. Henceforth we refer to these as 'small' and 'large' pits. We could not identify any evidence of a border in any of the pits found along the fibre cell walls. All fibres examined showed scarce simple pitting across the cell walls, often with both types of pits present. These findings contradict the earlier work of [32], who stated that pits along fibre tracheid cell walls are spread evenly along the length of the cell, while pits present on libriform fibres are concentrated towards the centres.

The small pits we observed were more circular, seeming to occur less frequently than the larger fibre pits and usually in closer proximity to other small pits. The larger pits were elliptical and stretched vertically. The average diameter of the small pits was 0.7 ± 0.2 μm while the larger pits measured 1.6 ± 0.4 μm and 1.1 ± 0.3 μm for the major and minor diameters, respectively. **Fig 8** shows the distribution of all pit sizes, which seem to occupy two overlapping population distribution, smaller pits with an average diameter around 0.6 μm and larger pits with an average diameter of around 1.3 μm.

The only fibre pitting measurement we were able to obtain from existing literature was from the work of Cirelli, Jagels [11] who reported only small and circular pits along libriform fibre walls measuring 0.78 ± 0.35 μm in diameter and on fibre tracheids bordered pits with elliptical apertures measuring 1.7 μm and 0.9 m for the major and minor axes respectively. We have manually looked over hundreds of sugar maple fibres and found both pit types (small and

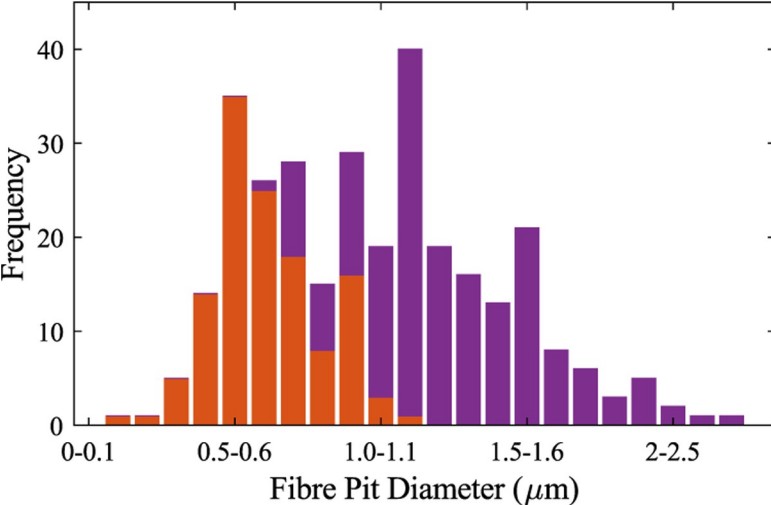

**Fig 8. Histogram of fibre pit diameters.** 308 SEM measurements are shown with small pits in orange and large pits in purple. Pits were assigned to either group based on shape.

large) exist on the same fibre for the majority of fibres observed. Our measurements of the small pits do compare well to the measurements of [11], as seen in Table 1.

As stated, we observed both large and small simple pits scattered across the walls of all fibres observed. As we examined hundreds of fibres, it seems unlikely that none were in contact with other cell types. This suggests that neighbouring cell type does not impact fibre pitting in young maples. We found some simple pits between fibres appeared blind, whilst some formed a simple pit pair **Fig 7(K)–7(M)**. It is possible that these simple pits were also connecting fibres and other xylem elements, indeed fibre tracheids are believed to be connected to vessels via bordered pits [11]. However, identifying neighbouring cell types from macerations is challenging, and so this is presented with caution.

The results of our examination of fibre pitting suggest that pitting is not an accurate way to differentiate fibre tracheids from libriform fibres in sugar maple stems, at least in young maple saplings. More work should be conducted to better understand fibre pitting. We believe the best place to start would be to adapt the differential staining technique, used to determine fibre types in transverse sections, to stain macerated fibre cells so fibres can be differentiated prior to examining the pitting. Further suggestions of additional experiments to differentiate fibre types in *Acer* come from the work of [43]. They found that using ultraviolet light (355–375 nm excitation) would provide an effective method to distinguish fibre types based on cell wall composition. Additionally, the microfibril angle for libriform fibres was significantly larger than those of fibre tracheids. They also identified that libriform fibres were more easily damaged, while fibre tracheids remained relatively intact. Finally, in sections of *Acer saccharum* that were exposed to *Ceriporiopsis subvermispora* (a white-rot fungus), areas of libriform fibres

**Table 1. Comparison between measured pitting sizes and the work of [11].**

|  | Type of Pit | Diameter | Major | Minor |
|---|---|---|---|---|
| Our Results | Small Pits | 0.66 μm |  |  |
| Cirelli, Jagels [11] | Libriform Fibre Pits | 0.78 μm |  |  |
| Our Results | Large Pits |  | 1.58 μm | 1.06 μm |
| Cirelli, Jagels [11] | Fibre Tracheid Pits |  | 1.7 μm | 0.9 μm |

were considerably more degraded than areas of fibre tracheids [41]. These findings also provide further evidence that libriform fibres are lignified differently than fibre tracheids [22].

## 4. Conclusion

In this paper we have extracted detailed information on the size, cell wall structure, pitting along cell walls and distribution of different cell types throughout the xylem. The results obtained were in general agreement with the few existing findings on mature sugar maple xylem, though the measured fibre and vessel element length was notably smaller than previously reported. These measurements provide valuable insight to build upon in future work.

Whilst we have obtained detailed information on different cell types, there remains a need to improve cell identification. Certain elements, such as vessels and ray parenchyma, are easily separated and are easily identified using different orientations of the xylem. However, we were unable to distinguish between living fibres and axial parenchyma, to fully distinguish living fibres from other fibre types, or to distinguish libriform fibres from fibre tracheids.

The inability to distinguish between libriform versus tracheid fibres was most notable. Existing literature suggests we should see distinct difference in pitting between the two types, but in practise we saw similar pitting across all fibres. That we saw similar pitting on all examined fibres suggests connections between fibres and other cell types, which likely plays a role in sugar maples unique sap exudation behaviour. However, it also means that pitting cannot be used to distinguish fibre types. The differential staining technique tested was promising in this regard, but requires further refinement and adaptation for application to macerated samples.

## Author Contributions

**Data curation:** Tenaya Driller.

**Formal analysis:** Tenaya Driller.

**Funding acquisition:** Matthew Watson.

**Investigation:** Tenaya Driller.

**Methodology:** Tenaya Driller.

**Supervision:** Matthew Watson.

**Visualization:** Tenaya Driller, James A. Robinson.

**Writing – original draft:** Tenaya Driller.

**Writing – review & editing:** James A. Robinson, Mike Clearwater, Daniel J. Holland, Abby van den Berg, Matthew Watson.

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
