## [Decision Letter · Decision Letter 0]

24 May 2023

PONE-D-23-05157Anatomy of the Juvenile Sugar Maple XylemPLOS ONE

Dear Dr. Robinson,

Thank you for submitting your manuscript to PLOS ONE. After careful consideration, we feel that it has merit but does not fully meet PLOS ONE’s publication criteria as it currently stands. Therefore, we invite you to submit a revised version of the manuscript that addresses the points raised during the review process.

We look forward to receiving your revised manuscript.

Kind regards,

Fabricio José Pereira, Ph.D.

Academic Editor

PLOS ONE

Journal Requirements:

3. We noted in your submission details that a portion of your manuscript may have been presented or published elsewhere. Please clarify whether this publication was peer-reviewed and formally published. If this work was previously peer-reviewed and published, in the cover letter please provide the reason that this work does not constitute dual publication and should be included in the current manuscript.

Reviewers' comments:

Reviewer's Responses to Questions

**Comments to the Author**

1. Is the manuscript technically sound, and do the data support the conclusions?

Reviewer #1: Partly

Reviewer #2: Yes

2. Has the statistical analysis been performed appropriately and rigorously? 

Reviewer #1: No

Reviewer #2: Yes

3. Have the authors made all data underlying the findings in their manuscript fully available?

Reviewer #1: No

Reviewer #2: No

4. Is the manuscript presented in an intelligible fashion and written in standard English?

Reviewer #1: No

Reviewer #2: Yes

5. Review Comments to the Author

Reviewer #1: Complete Scientific name with authors and botanical family; In this first paragraph of the introduction the authors present the species, being able to add the correct scientific name according to the nomenclature, with authors and family; There´s more study about this? To complete introduction; Botanical identification of the study species?

How many samples do you guys have? Threes?;Paper follow the comments IAWA commits? there is no citation.

The imagens without orientation vertical; The authors use different histochemical staining methods but place the image in black and white; The authors do not comment qualitative aspect like parenchyma axial type, and others usually in wood anatomy study; So, wood anatomy is set of macroscopic and microscopic anatomical features qualitative and quantitative; Graphics with frequency and length of vessel elements;I think the article has potential for publication as long as the comments are observed; Review the bibliographic references because there are references that are not mentioned in the text; Review guidelines; I recommend changing the title of the work to something around quantitative anatomy, or quantitative analyzes of the anatomical structure of wood, something like that. Wood anatomy is more ample.

Reviewer #2: This paper is a great summary of anatomical properties of young sugar maple, which will be very useful to improve our models of maple sap exudation. The paper is an easy-to-follow and generally clearly written summary of the anatomical properties derived from images analysis. The methods are sound and the conclusion well supported. I only a have a few minor comments and request for changes to make sure that the paper meets reproducibility standards and is clear. I also add detailed comments (line-by-line) for the authors, which they might want to consider to make the paper more useful for the reader. Overall, I aplaud the authors for their thourough work that acknowledges its caveats and limitations openly. It was a pleasure reviewing this and I apologies that I could not get the review done quicker.

Comments and requests by line:

L23f: While the value observed here are generally in agreement with previous studies, there are also a few measurements that are different and the authors speculate that this may be related to tree size later in the text. I think it would be nice to highlight these potential differences between young and mature tree here for the reader.

L27f: Sap exudation happen at atmospheric pressure or lower, notably when vacuum tubing is used. Given that the author use a method of sap extraction under vacuum for young maples as a justification for their work, they may want to highlight the fact that vacuum can be applied to increase sap extraction when talking about atmospheric pressure.

L28ff: For your information, I would and have argued with co-authors (Rademacher et al., 2023) that the mechanism is well constrained, albeit it not completely resolved.

L30f: This sentence is imprecise and needs to be revised. In particular I have two issues here. First, the xylem is not the only region of sap conduction. In fact, the phloem is much more important for solute transport. Second and more importantly, referring to storage of phosynthates is very imprecise here. The xylem stores nonstructural carbohydrates, but they have been transported and transformed multiple times before and are not really photsynthates anymore. Please revise the sentence to be more precise.

L33: This ability to develop elevated stem pressure in a leafless state is not unique to Acer. At least, some Betula, Juglans, and other species are also able to develop positive stem pressures in a leafless state, although the mechanism is different for some of these species (Hölltä et al., 2018).

L39: Could be shortened to "capillary and osmotic forces".

L48: There are also the following papers with relevant measurements that ought to be cited here: Ellmore et al (2006), Gregory (1977), Gregory (1978), Wallner & Gregory (1980), and Wason et al. (2019).

L51: The authors should use SI units, aka cm here.

L51ff: This may be true but sap volume yield is exponentially related to tree size (Rademacher et al., 2023). It would be interesting to calculate yield on a per area instead of an per tap basis to better compare different types of maple groves.

L67f: How many saplings were sampled, from which tissue did you sample, what were the dimensions of the samples, and how many samples were prepared, imaged, and analysed using each method? This information is crucial to better understand the within- and between-tree variability and also needs to be added for the purpose of reproducibility and to be able to interpret the reported results with regard to the sample size.

L72: I assume the authors mean paraffin when referring to wax, please change this to be more specific.

L75f: It would be interesting to know why you used two different staining techniques and for reproducibility, you need to specify when which technique was used.

L88: Please specify the threshold used here for reproducibility.

L90: Please change "the standard deviation is used as the error." to "one standard deviation is reported as a measure of the variability."

L114: How can the mode be a range of numbers. It should just be a single integer. Either the sentence is not clear or there is an error here.

L119: Typo "ray" instead of "rays".

L120: This is the first time the reader hears about the dimensions of the samples. This type of information needs to be included in the methods section, as I already mentionned above.

L125: Typo "rays" instead of "ray".

L125: Surely, the distance only remained roughly the same and not exactly.

L126: Between each ray or between rays?

L128: The sentence is missing a preposition. Do you mean "identified by a highly lignified cell wall"?

L129 & L133: I would prefer if you made it clear that you are talking about the genus here.

L140ff: It is not clear why this aspect of sugar maple anatomy requires further investigation. Please speel out to the reader, why this is the case.

L142: Personally, I would switch the last two sentences of the paragraph, as the final sentence follows nicer from the third to last sentence.

L146: This is relative to other vessels, as the are all wide compared to fibres, right? Mabybe make this clear.

L160: I am not sure facilitating is the right word here.

L167: Again, how can the mode be several values. Are you maybe talking about the range of the modes from each sample? Something is not clear here and needs to be revised. Same as comment L114.

L172: There is also the more recent work by Savage et al. (2017) that shows clear within-tree variation of phloem conduit diameter for red maple.

L174: Give us the sample size of observations here. This can be powerful (n >100) and anecdotal (n=3) depending on sample size.

L206: Why not give the mean and one standard deviation here, as done elsewhere?

L211: Maybe "measurements" or "observations" would be preferable to "experiment" in this line?

L216 & L218: Why the parenthesis in the figure reference. You did not use parenthesis before. Why not stay consistent?

L223f: What is this believe based on? Could that evidence that this techniques is promising be presented? Maybe add some numbers or figures that highlight, why the authors think this method is promissing?

L224: Don't you mean "fibre types", rather than "fibre groupings"?

L251: Typio "cell wall".

L255f: This sentence is not as helpful as it coule be. The sentence implies that they did find some, but does not state it. Did they observed axial parenchyma-fibre pits or not in this previous study? Please rephrase to clarify.

L269: Maybe replace ", perfect for investigating" with "to measure", as it is clearly not perfect given the following statement.

L273: It seems strange to say: We don't know what was next to it, but the pitting was dependent on what was next to it. Just changing it to "Vessel pitting seems to be extremely ..." would work better in my opinion.

L293: "Vessels"seems more appropriate here than the "tree stem's vascular system" here.

L298f: Please elaborate what you mean by the different roles are apparent from the pitting.

L323f: Everything following the comma is repetitive and redundant and could be cut.

L338: You mean to distinct types of pits? You found plenty of distinct pits, but I think you refer to the two sub-types of pits in fibres, that are discussed here.

L339: It would be clearer to start the sentence with "Henceforth," or a similar expression.

L348fff: It would be more useful for the reader to separate the distribution for the two types of pits. This could be done by simply using two different colours, as in the other histograms. I find it hard to se the bi-model distribution that would result from two overlapping normal distributions, but another representation might be helpful to make this clearer to me and other readers.

L362: Please add "in young maples" to this sentence.

L369: Again, please add a reference to the yougn age of the trees examined here.

L384: This is true, but I would also highlight the difference that they did find.

L395: All fibres?

L399ff: I would cut this, as it feels redundant. There is always more work that could be done and there are already more studies on mature than on juvenile maples.

References

Ellmore et al. (2006) "Comparative sectoriality in temperate hardwoods: hydraulics and xylem anatomy", Botanical Journal of the Linnean Society, 150(1), 61-71

Gregory (1977) "Cambial activity and ray cell abundance in Acer saccharum", Canadian Journal of Botany, 55(20), 2559-2564

Gregory (1978) "Living elements of the conducting secondary xylem of sugar maple (Acer saccharum Marsh.)", 4, 65-69

Hölttä et al. (2018) "Water relations in silver birch during springtime: How is sap pressurised?", Plant Biol., 20, 834–847

Rademacher et al. (2023) "TAMM review: On the importance of tap and tree characteristics in maple sugaring", Forest Ecology and Management, 535, 120896

Savage et al. (2017) "Maintenance of carbohydrate transport in tall trees", 3(12), 965-972, doi: 10.1038/s41477-017-0064-y

Wallner & Gregory (1980) "Relationship of sap sugar concentrations in sugar maple to ray tissue and parenchyma flecks caused by Phytobia setosa", Canadian Journal of Forest Research, 10, 312-315

Wason et al. (2019) "The functional implications of tracheary connections across growth rings in four northern hardwood trees", 124(2), 297-306, doi: 10.1093/aob/mcz076

Best wishes,

Tim Rademacher

Researcher at the Centre ACER

Associate Professor at the Université du Québec en Outaouais

6. PLOS authors have the option to publish the peer review history of their article (what does this mean?). If published, this will include your full peer review and any attached files.

Reviewer #1: No

Reviewer #2: **Yes: **Tim Rademacher

---

## [Author Response · Author response to Decision Letter 0]

3 Jul 2023

We thank the reviewers for their feedback. We have separated our response to each reviewers’ comments below.

Reviewer #1:

-We have endeavoured to respond to the individual comments made, where possible, though for several we were unsure if comments were related or meant to be entirely separate.

"Reviewer #1: Complete Scientific name with authors and botanical family; In this first paragraph of the introduction the authors present the species, being able to add the correct scientific name according to the nomenclature, with authors and family;"

-We have added the botanical authority to the first mention of the binomial species name in the introduction (see line 28).

"There´s more study about this? To complete introduction; Botanical identification of the study species?"

-It is not clear what the reviewer is asking for here. In response to the previous comment, we have added the authority to the first mention of the species name in the introduction. Elsewhere in the manuscript the common name is used. Sugar maple is a well known species, with no confusion around use of the common or binomial name, we do not feel that further botanical description will add to the manuscript.

"How many samples do you guys have? Threes?; Paper follow the comments IAWA commits? there is no citation."

-In our work we took multiple samples (the exact no. was not recorded) of branch and stem sections 3-15 mm diameter off of 5 saplings. We have modified the methods section to include this additional detail (see line 71-76).

"The imagens without orientation vertical;"

-As requested we have gone through the images and aligned them as consistently as possible, whist ensuring desired features are still visible.

"The authors use different histochemical staining methods but place the image in black and white;"

-As requested, all black and white LOM images have been replaced by colour versions.

"The authors do not comment qualitative aspect like parenchyma axial type, and others usually in wood anatomy study; So, wood anatomy is set of macroscopic and microscopic anatomical features qualitative and quantitative;"

-Our work is primarily focussed on xylem cells which are known to contribute to winter pressure generation in sugar maple, in particular the fibres and vessel elements. For this reason, along with our inability to reliably distinguish axial parenchyma from other xylem cell types, we do not examine axial parenchyma in depth in this work

"Graphics with frequency and length of vessel elements;"

-The frequency and length of vessel elements was shown in Figure 5a. We have adjusted the axis labels and figure caption to make this clearer.

"Review the bibliographic references because there are references that are not mentioned in the text;"

-Assuming the reviewer is talking about our references in the bibliography, we have checked and can confirm all references in the bibliography are cited in the text.

"Review guidelines; I recommend changing the title of the work to something around quantitative anatomy, or quantitative analyzes of the anatomical structure of wood, something like that. Wood anatomy is more ample."

-As requested, we have changed the title of the manuscript to “Quantitative Examination of the Anatomy of the Juvenile Sugar Maple Xylem”

 

Reviewer #2:

“Reviewer #2: This paper is a great summary of anatomical properties of young sugar maple, which will be very useful to improve our models of maple sap exudation. The paper is an easy-to-follow and generally clearly written summary of the anatomical properties derived from images analysis. The methods are sound and the conclusion well supported. I only a have a few minor comments and request for changes to make sure that the paper meets reproducibility standards and is clear. I also add detailed comments (line-by-line) for the authors, which they might want to consider to make the paper more useful for the reader. Overall, I aplaud the authors for their thourough work that acknowledges its caveats and limitations openly. It was a pleasure reviewing this and I apologies that I could not get the review done quicker.”

-We have responded to their individual comments and made modifications as outlined below:

-For the following comments we have modified the text as suggested by the reviewer, or made simple changes which should account for the minor issues listed:

“L39: Could be shortened to "capillary and osmotic forces".”

“L51: The authors should use SI units, aka cm here.”

“L72: I assume the authors mean paraffin when referring to wax, please change this to be more specific.”

“L90: Please change "the standard deviation is used as the error." to "one standard deviation is reported as a measure of the variability."”

“L119: Typo "ray" instead of "rays".”

“L125: Typo "rays" instead of "ray".”

“L125: Surely, the distance only remained roughly the same and not exactly.”

“L126: Between each ray or between rays?”

“L128: The sentence is missing a preposition. Do you mean "identified by a highly lignified cell wall"?”

“L129 & L133: I would prefer if you made it clear that you are talking about the genus here.”

“L142: Personally, I would switch the last two sentences of the paragraph, as the final sentence follows nicer from the third to last sentence.”

“L146: This is relative to other vessels, as the are all wide compared to fibres, right? Mabybe make this clear.”

“L211: Maybe "measurements" or "observations" would be preferable to "experiment" in this line?”

“L216 & L218: Why the parenthesis in the figure reference. You did not use parenthesis before. Why not stay consistent?”

“L251: Typio "cell wall".”

“L269: Maybe replace ", perfect for investigating" with "to measure", as it is clearly not perfect given the following statement.”

“L273: It seems strange to say: We don't know what was next to it, but the pitting was dependent on what was next to it. Just changing it to "Vessel pitting seems to be extremely ..." would work better in my opinion.”

“L293: "Vessels"seems more appropriate here than the "tree stem's vascular system" here.”

“L323f: Everything following the comma is repetitive and redundant and could be cut.”

“L338: You mean to distinct types of pits? You found plenty of distinct pits, but I think you refer to the two sub-types of pits in fibres, that are discussed here.”

“L339: It would be clearer to start the sentence with "Henceforth," or a similar expression.”

“L362: Please add "in young maples" to this sentence.”

“L369: Again, please add a reference to the yougn age of the trees examined here.”

“L395: All fibres?”

“L399ff: I would cut this, as it feels redundant. There is always more work that could be done and there are already more studies on mature than on juvenile maples.”

-These comments required more detailed or considered modification and so we have responded individually (though some comments concerning similar issues have been grouped):

“L27f: Sap exudation happen at atmospheric pressure or lower, notably when vacuum tubing is used. Given that the author use a method of sap extraction under vacuum for young maples as a justification for their work, they may want to highlight the fact that vacuum can be applied to increase sap extraction when talking about atmospheric pressure.”

-We have added reference to the common use of vacuum in maple sap collection (see lines 30-31).

“L28ff: For your information, I would and have argued with co-authors (Rademacher et al., 2023) that the mechanism is well constrained, albeit it not completely resolved.”

-The reviewer’s comment here is interesting. The cited work notes the papers by Stockie (and co) when stating the mechanism is well constrained. Stockie’s model is a promising demonstration that the mechanism outlined by Tyree (1995) can explain sap exudation. However, as the reviewer notes, the mechanism is still not completely resolved. Our research group has also been involved in work (yet to be published) using microCT to further validate the assumptions underlying this mechanism.

-We have modified the text slightly to capture the reviewer’s point (see lines 31-32).

“L30f: This sentence is imprecise and needs to be revised. In particular I have two issues here. First, the xylem is not the only region of sap conduction. In fact, the phloem is much more important for solute transport. Second and more importantly, referring to storage of phosynthates is very imprecise here. The xylem stores nonstructural carbohydrates, but they have been transported and transformed multiple times before and are not really photsynthates anymore. Please revise the sentence to be more precise.”

-We have revised the sentence as requested (see lines 33-35).

“L33: This ability to develop elevated stem pressure in a leafless state is not unique to Acer. At least, some Betula, Juglans, and other species are also able to develop positive stem pressures in a leafless state, although the mechanism is different for some of these species (Hölltä et al., 2018).”

-The reviewer is correct that while it’s mechanism for developing elevated stem pressures differs from many other species, it is not “unique”. We have removed the word in question from that sentence.

“L48: There are also the following papers with relevant measurements that ought to be cited here: Ellmore et al (2006), Gregory (1977), Gregory (1978), Wallner & Gregory (1980), and Wason et al. (2019).”

-As requested, we have added references to all recommended, except Gregory 1978. This paper is cited later in the text, and while the work does look at anatomy it does not attempt to provide any direct measurements of xylem features, unlike the other suggested papers. 

“L51ff: This may be true but sap volume yield is exponentially related to tree size (Rademacher et al., 2023). It would be interesting to calculate yield on a per area instead of an per tap basis to better compare different types of maple groves.”

-The reviewer’s point is interesting in consideration with the benefits to the increased density of the plantation method. We are currently in the process of growing saplings to test the sap collection approach. Once the saplings are large enough for harvest, this would be an interesting metric to evaluate.

“L67f: How many saplings were sampled, from which tissue did you sample, what were the dimensions of the samples, and how many samples were prepared, imaged, and analysed using each method? This information is crucial to better understand the within- and between-tree variability and also needs to be added for the purpose of reproducibility and to be able to interpret the reported results with regard to the sample size.”

“L120: This is the first time the reader hears about the dimensions of the samples. This type of information needs to be included in the methods section, as I already mentioned above.”

-As requested, we have added additional detail to the methods section outlining how many saplings were analysed in this work (see lines 71-76). We have added additional detail on how many different images were analysed (see lines 96-97). Unfortunately the number of samples taken from each sapling was not recorded and so cannot be included. 

“L174: Give us the sample size of observations here. This can be powerful (n >100) and anecdotal (n=3) depending on sample size.”

-The number of vessels on which spiral thickening was observed was not recorded. However, we have modified the text to include a general estimate as well as clarified how many vessel elements were directly analysed (see lines 185-188).

“L75f: It would be interesting to know why you used two different staining techniques and for reproducibility, you need to specify when which technique was used.”

-Of the two staining techniques implemented, the first (safranin and fast green) was used as a standard stain for highlighting general features when viewing under the microscope. All figures, unless otherwise stated (i.e. the caption of Figure 3 for images h-i), used this stain.

-The second stain, safranin and astra blue, was used to try and stain different fibre types in different colors based on lignin content. The results of this staining are discussed in depth in section 3.A.IV.

-Analysis of cell size was done on samples stained with either staining, since generally the choice of staining did not impact our ability to measure desired features from images. Additionally, we used a third staining approach for some images, that being staining with just Safranin. This was used, as with safranin and fast green, for highlighting basic features when viewed under the microscope.

-We have modified the text to clarify things on lines 81-85.

“L88: Please specify the threshold used here for reproducibility.”

As each set of images taken will vary slightly, (i.e. for LOM images minor variations in staining or slice thickness will affect the image) it was necessary to use a different threshold for each image set. This threshold was manually chosen based on what best segmented the fibres and vessels and so an exact value cannot be provided. We have clarified this point in the text (lines 98-99).

“L114: How can the mode be a range of numbers. It should just be a single integer. Either the sentence is not clear or there is an error here.”

“L167: Again, how can the mode be several values. Are you maybe talking about the range of the modes from each sample? Something is not clear here and needs to be revised. Same as comment L114.”

-We thank the reviewer for this comment. Our wording here was imprecise. We were referring to the highest frequency range of the associated histogram (i.e. the range on the histogram into which the largest number of values fall). We have modified the text to clarify this point on lines 125-126 and 177-178).

“L140ff: It is not clear why this aspect of sugar maple anatomy requires further investigation. Please spell out to the reader, why this is the case.”

-There is currently no obvious way to reliably distinguish axial parenchyma and living fibres when viewed under SEM/LOM, if we wanted to provide anatomical measurements of these features a way of reliably distinguishing the two would need to be developed. We have clarified this point in the text on lines 151-153.

“L160: I am not sure facilitating is the right word here.”

-We have reworded the sentence to remove the use of ‘faciliating (lines 169-171). We have made similar modifications throughout the paper to remove similar uses of facilitate/facilitating in the context of pit connections.

“L172: There is also the more recent work by Savage et al. (2017) that shows clear within-tree variation of phloem conduit diameter for red maple.”

-We have added a reference to the noted paper.

“L206: Why not give the mean and one standard deviation here, as done elsewhere?”

-This line is actually referring to the mode of the binned histogram data for the fibre diameter shown in Figure 6b. In this case, the line was deemed unnecessary, and has been removed: 

“L223f: What is this believe based on? Could that evidence that this techniques is promising be presented? Maybe add some numbers or figures that highlight, why the authors think this method is promising?”

“L224: Don't you mean "fibre types", rather than "fibre groupings"?”

-Using this technique, we have stained fibres different colors, which following Vazques-cooz and Meyer (2002) would suggest they are different fibre types. However we were unable to achieve the same level of clear distinction between regions as in Vazques-cooz and Meyer (2002). Therefore although the technique is promising, the method needs further refinement.

-The reviewer is also correct that fibre ‘types’ is more accurate. We used groupings as different types of fibres are expected to appear in ‘groups’ surrounded by similar fibres that all stain the same color.

-We have modified the wording in this paragraph to improve clarity (lines 231-236).

“L255f: This sentence is not as helpful as it could be. The sentence implies that they did find some, but does not state it. Did they observed axial parenchyma-fibre pits or not in this previous study? Please rephrase to clarify.”

-The referenced study did not observe any pits between fibres and axial parenchyma. We have reworded the sentence to make this clear (see lines 267-268).

“L298f: Please elaborate what you mean by the different roles are apparent from the pitting.”

-The line in question is confusing and we have chosen to delete it.

“L348fff: It would be more useful for the reader to separate the distribution for the two types of pits. This could be done by simply using two different colours, as in the other histograms. I find it hard to se the bi-model distribution that would result from two overlapping normal distributions, but another representation might be helpful to make this clearer to me and other readers.”

-As recommended we have modified Figure 8 to show the two pit types. The pits identified as small follow a normal distribution while the large pits follow a distribution that is nearly uniform at the higher end.

“L23f: While the value observed here are generally in agreement with previous studies, there are also a few measurements that are different and the authors speculate that this may be related to tree size later in the text. I think it would be nice to highlight these potential differences between young and mature tree here for the reader.”

“L384: This is true, but I would also highlight the difference that they did find.”

-We have modified the text in the abstract and conclusion in order to note the primary differences (see lines 24-26 and 395-396).

---

## [Decision Letter · Decision Letter 1]

25 Sep 2023

Quantitative Examination of the Anatomy of the Juvenile Sugar Maple Xylem

PONE-D-23-05157R1

Dear Dr. Robinson,

We’re pleased to inform you that your manuscript has been judged scientifically suitable for publication and will be formally accepted for publication once it meets all outstanding technical requirements.

Kind regards,

Fabricio José Pereira, Ph.D.

Academic Editor

PLOS ONE

Additional Editor Comments (optional):

Reviewers' comments:

Reviewer's Responses to Questions

**Comments to the Author**

1. If the authors have adequately addressed your comments raised in a previous round of review and you feel that this manuscript is now acceptable for publication, you may indicate that here to bypass the “Comments to the Author” section, enter your conflict of interest statement in the “Confidential to Editor” section, and submit your "Accept" recommendation.

Reviewer #2: All comments have been addressed

2. Is the manuscript technically sound, and do the data support the conclusions?

Reviewer #2: Yes

3. Has the statistical analysis been performed appropriately and rigorously? 

Reviewer #2: Yes

4. Have the authors made all data underlying the findings in their manuscript fully available?

Reviewer #2: No

5. Is the manuscript presented in an intelligible fashion and written in standard English?

Reviewer #2: Yes

6. Review Comments to the Author

Reviewer #2: The authors did a great job in revising the manuscript and addressed all of my concerns. I feel the manuscript is now ready for publication.

7. PLOS authors have the option to publish the peer review history of their article (what does this mean?). If published, this will include your full peer review and any attached files.

Reviewer #2: **Yes: **Tim Rademacher

---

## [Editor Report · Acceptance letter]

3 Oct 2023

PONE-D-23-05157R1 

Quantitative Examination of the Anatomy of the Juvenile Sugar Maple Xylem 

Dear Dr. Robinson:

I'm pleased to inform you that your manuscript has been deemed suitable for publication in PLOS ONE. Congratulations! Your manuscript is now with our production department. 

Kind regards, 

on behalf of

Dr. Fabricio José Pereira 

Academic Editor

PLOS ONE